# Mixed Effect of Alcohol, Smoking, and Smokeless Tobacco Use on Hypertension among Adult Population in India: A Nationally Representative Cross-Sectional Study

**DOI:** 10.3390/ijerph19063239

**Published:** 2022-03-09

**Authors:** Prashant Kumar Singh, Ritam Dubey, Lucky Singh, Nishikant Singh, Chandan Kumar, Shekhar Kashyap, Sankaran Venkata Subramanian, Shalini Singh

**Affiliations:** 1Division of Preventive Oncology & Population Health, ICMR-National Institute of Cancer Prevention and Research, Noida 201301, India; ritam.dubey@gmail.com (R.D.); nishiiips@gmail.com (N.S.); 2WHO FCTC Knowledge Hub on Smokeless Tobacco, ICMR-National Institute of Cancer Prevention and Research, Noida 201301, India; shalinisingh.icmr@gmail.com; 3ICMR-National Institute of Medical Statistics, New Delhi 110029, India; lucky.5bhu@gmail.com; 4Department of Policy and Management Studies, TERI School of Advanced Studies, New Delhi 110070, India; c.kumar803@gmail.com; 5Department of Cardiology, Army Research & Referral Hospital, New Delhi 110010, India; shekharkashyap@gmail.com; 6Harvard Center for Population and Development Studies, Cambridge, MA 02138, USA; svsubram@hsph.harvard.edu; 7Department of Social and Behavioral Sciences, Harvard T.H. Chan School of Public Health, Boston, MA 02115, USA; 8ICMR-National Institute of Cancer Prevention and Research, Noida 201301, India

**Keywords:** hypertension, alcohol, smokeless tobacco, smoking, India

## Abstract

Sporadic evidence is available on the association of consuming multiple substances with the risk of hypertension among adults in India where there is a substantial rise in cases. This study assesses the mutually exclusive and mixed consumption patterns of alcohol, tobacco smoking and smokeless tobacco use and their association with hypertension among the adult population in India. Nationally representative samples of men and women drawn from the National Family and Health Survey (2015–2016) were analyzed. A clinical blood pressure measurement above 140 mmHg (systolic blood pressure) and 90 mmHg (diastolic blood pressure) was considered in the study as hypertension. Association between mutually exclusive categories of alcohol, tobacco smoking and smokeless tobacco and hypertension were examined using multivariate binary logistic regression models. Daily consumption of alcohol among male smokeless tobacco users had the highest likelihood to be hypertensive (OR: 2.32, 95% CI: 1.99–2.71) compared to the no-substance-users. Women who smoked, and those who used any smokeless tobacco with a daily intake of alcohol had 71% (OR: 1.71, 95% CI: 1.14–2.56) and 51% (OR: 1.51, 95% CI: 1.25–1.82) higher probability of being hypertensive compared to the no-substance-users, respectively. In order to curb the burden of hypertension among the population, there is a need for an integrated and more focused intervention addressing the consumption behavior of alcohol and tobacco.

## 1. Introduction

Hypertension or high blood pressure (High BP) has been a growing public health concern in low- and middle-income countries [1]. A major risk factor for cardiovascular, cerebrovascular and kidney diseases [2], hypertension has been extensively associated with alcohol and tobacco consumption. Tobacco smoking has been associated with lowered blood pressure among males [3] and high blood pressure among females [4]. Smokeless tobacco (SLT) has been associated with elevated blood pressure among both the sexes of the Indian subcontinent [5]. The sex differentials in tobacco consumption have been widely observed in the South-Asian region. The consumption of both forms of tobacco (smoking and smokeless) has been higher among men than women, while smokeless tobacco has a more pervasive reach across the social strata [6]. According to recent nationally representative surveys, 26.7% scores of adults were noted to consume tobacco in India around 2016–2017, out of which about 8% were known to use both smokeless and smoking forms [7,8].

Another notable substance known to raise blood pressure promptly is alcohol [9]. Around 2.6% prevalence of alcohol use disorder was observed in the country [10]. The health repercussions of simultaneous consumption of tobacco and alcohol as well as the interdependence in consumption of both the substances are broadly noted in the literature [11,12]. However, there is little evidence on the association of their combined consumption with hypertension [13].

The literature also lacks exploration accounting for the quantity of intake and type of product consumed for both the substances and their association with hypertension. The relevance of exploring the precipitating effects of the exclusive and combined use of substances on related health outcomes is pertinent to help develop a comprehensive and multidimensional evidence pool to address the increasing prevalence of non-communicable diseases in countries, such as India. The intake of alcohol and tobacco among the Indian population is considerably higher, and in a few regions, it is also a culturally well-accepted practice [14]. In addition to achieving the ambitious target of reducing premature mortality by non-communicable diseases to a one-third set by the Sustainable Development Goal-3 [15], the urgency of gaining insight has increased manifold within the context of the COVID-19 pandemic. There is adequate evidence to demonstrate an increased risk of hospitalization and mortality among hypertensive patients contracting the COVID-19 infection [16].

The present study incorporates a sex-stratified analysis to evaluate associations of exclusive and mixed substance use—alcohol, smoking tobacco, and smokeless tobacco—with the risk of the hypertensive condition among the adult population, using nationally representative samples from India. This study used the conceptual framework on the social determinants of health (CSDH) proposed by the World Health Organization (WHO) for selecting social, demographic, economic and regional correlates to be adjusted in the analyses [17].

## 2. Materials and Methods

### 2.1. Data Source and Study Samples

This study uses the nationally representative and cross-sectional survey data collected in the fourth round of the National Family Health Survey 2015–2016 (NFHS-4) [18]. The NFHS aims to collect essential data on health and family welfare as well as on related emerging issues. The clinical, anthropometric and biochemical components of the survey provides information to estimate the prevalence for a range of nutritional and lifestyle ailments through a series of biomarker tests and measurements for women aged 15–49 years and men aged 15–54 years. The NFHS-4 uses a two-stage stratified sampling design covering urban and rural areas of 640 districts, 29 states, and seven union territories (UTs). There were a total of 601,509 households with a 98% response rate, where 699,686 women aged 15–49 years and 112,122 men aged 15–54 were interviewed, covering 97% and 92% response rates, respectively. Detailed information about the sampling frame, survey design, and survey instruments used in NFHS-4 can be accessed at http://rchiips.org/nfhs/NFHS-4Report.shtml.

The study samples included 101,408 males and 678,460 females aged 15–49 years, excluding the participants who reported on anti-hypertensive medications (Figure 1).

### 2.2. Ethical Approval

The current study undertook no direct recruitment of participants, rather, it utilized the publicly available NFHS-4 dataset, which conforms to the principles embodied in the Declaration of Helsinki. The dataset does not include the identifiers of the individuals involved in the study. The participants extended their oral consent before administering any questionnaire or biomarker test in the survey. The ethical review board of the International Institute for Population Sciences (IIPS), Mumbai, India, granted ethical approval to the protocol and survey instruments of the NFHS-4. The ethical review procedures and the survey tools were also reviewed and approved by the ICF International Review Board.

### 2.3. Measures

#### 2.3.1. Hypertension

A sample participant was considered hypertensive in the present study when the average measurement of systolic blood pressure (SBP) and the diastolic blood pressure (DBP) was more than 140 mmHg and 90 mmHg, respectively. Blood pressure was measured in the survey using the Omron Blood Pressure Monitor. Three readings of blood pressure were taken with an interval of 5 min from each eligible male and female participant after obtaining their consent. Participants who reported to be on anti-hypertensive medications were excluded from the study to control for the pharmacological effects on the observations included in the analysis.

#### 2.3.2. Mixed Combination of Substance Use

This study considered the mixed consumption pattern of alcohol and tobacco as the key predictor variable to associate with the hypertensive status of individuals. The survey provides information on the current use and frequency of intake of alcohol, smoking and smokeless tobacco. Using the survey information, a composite variable was generated, representing the mutually exclusive and combined/mixed consumption pattern of the three substances, i.e.,

(i)No substance use;(ii)Smokeless tobacco only;(iii)Tobacco smoking only;(iv)Smokeless tobacco and smoking tobacco;(v)Daily alcohol only;(vi)Daily alcohol and smokeless tobacco only;(vii)Daily alcohol and tobacco smoking only;(viii)Daily alcohol and both forms of tobacco;(ix)Irregular (not daily) alcohol only;(x)Irregular (not daily) alcohol and smokeless tobacco only;(xi)Irregular (not daily) alcohol and tobacco smoking only;(xii)Irregular (not daily) alcohol and both forms of tobacco.

Self-report of alcohol intake was noted for the individuals asking the question “Do you drink alcohol?” where the responses were sought in terms of “Yes” and “No”. Individuals responding “No” to the question were included under the category of “No alcohol intake”. A follow-up question, “How often do you drink alcohol?” was asked in the survey to gauge the frequency of alcohol intake in terms of “almost every day”, “about once a week”, or “less than once a week”. The study considered the frequency of alcohol consumption in a binary form where the responses to “almost every day” were categorized under the “daily alcohol intake” and “about once a week” and “less than once a week” were combined to be represented under “irregular (not daily) alcohol intake”.

Tobacco smoking was determined in the survey by asking eligible men and women, “Do you currently smoke cigarettes/bidis?” and participants responding “Yes” were considered under “tobacco smoking”. The intake of SLT was noted by asking participants, “In what other form do you currently smoke or use tobacco?” and the responses under the categories of “Gutkha/Paan masala with tobacco”, “Khaini”, “Paan with tobacco”, “Other chewing tobacco”, and “Snuff” were considered under the category of “smokeless tobacco”. The reference period for collecting the information regarding tobacco consumption from the individuals was the last 24 h preceding the date of the survey.

#### 2.3.3. Covariates

The study used select social, demographic, economic, and regional correlates of hypertension, guided by the WHO’s CSDH [17] and the literature. A range of individual social and demographic characteristics, including age, education level, marital status, occupation, and body mass index of the individuals, were included in the analyses. Studies have highlighted the significant association of these characteristics with health behaviors and resulting outcomes, such as chronic morbidities, e.g., hypertension [19,20]. The age was classified into seven groups (15–19, 20–24, 25–29, 30–34, 35–39, 40–44, and 45–49 years). Education level was categorized in accordance with the highest level of formal education completed by the individual, which is primary (from grade 1st to 5th), secondary (from 6th to 12th), and higher (graduate and above). Marital status was considered whether the individual was currently married, widowed/divorced/separated, not married, or never in a union. Occupational status was grouped into seven categories: unemployed individuals were included in ‘not in workforce’ where individuals involved in the varied mode of employment were categorized accordingly, such as manual-skilled and unskilled, services/household and domestic, agriculture, sales, clerical, and professional/technical/managerial. Body mass index (BMI) was computed as per the WHO’s standard norms for BMI classification, especially for the Asia-Pacific region [21] in terms of weight in kilograms divided by height in meters squared. The present study considered BMI in four categories that are normal (<18.5), underweight (18.5–24.9), overweight (25.0–29.9), and obese (≥30.0). The anthropometric data were considered for women who were not currently pregnant or gave birth in the two months preceding the survey date.

Socio-cultural and economic characteristics are noted in the literature to have a pertinent impact on health status and behaviors [22]. The survey provides aggregated information on these characteristics at the household level, which is common for all the members of the household. The economic status of a household is determined by the scores of a composite index based on household possession of vital resources of sustenance. They include the quality of the house, water, and sanitation facilities, and the ownership of the assets, such as house, computer, mobile phone, car, etc. This composite index is termed as ‘wealth index’ in demographic and health survey datasets, which are classified into five equal parts, i.e., quintile (and thus known as ‘wealth quintile’), based on the weighted scores of the index representing the range of household economic status from the poorest to the richest. Similarly, the household affiliation to religious and social groups was determined in the survey. The religious affiliation of household members was grouped into ‘Hindu’, ‘Muslim’, and ‘Others’, while the affiliation to social groups was classified into ‘Other Backward Class (OBC)’, ‘Scheduled Castes (SCs)’, ‘Scheduled Tribes (STs)’, and ‘Others’. The Government of India recognized a collective group of educationally or socially disadvantaged castes as OBC in 1980s. The SCs and STs are officially designated as the most disadvantaged socioeconomic groups of people in India. Articles 340, 341, and 342 of Indian Constitution specify the procedure to recognize OBC, SCs, and STs, respectively.

The literature has duly noted the association of geographical attribution and disparity with healthcare status and health outcomes, especially relevant for the diverse and densely populated country of India [23]. Place of residence (classified into rural and urban areas) and the region of residence (states) were also included as covariates in the analyses in order to adjust the regional disparity in assessing the association between substance use and the hypertensive status among the individuals.

### 2.4. Statistical Analysis

The prevalence of hypertension and its bivariate association with the combined/mixed substance use patterns and other covariates was assessed for both men and women aged 15–49 years separately, considering appropriate sampling weight and survey design. The Chi-squared tests were performed to explore the statistical significance of the bivariate association.

Binary logistic regression was used owing to the dichotomous nature of the outcome variable (hypertensive status), and the association of included explanatory variables with hypertension was assessed using the following logit link function;
$$\mathrm{Log}\left(\frac{p_{i}}{1-p_{i}}\right)=\beta_{0}+\beta_{1}X_{1}+\beta_{2}X_{2}\ldots .\beta_{k}X_{k}\;$$
where $p_{i}$ is the likelihood of an individual being hypertensive, $\beta_{0}$ is the log odds of the intercept, and $\beta_{1}$, …, $\beta_{k}$ are the coefficient for their corresponding predictor variables ($X_{1}$, …, $X_{k}$).

Adjusted odds ratios (ORs) with their respective 95% confidence intervals were calculated. All estimates were obtained using complex survey weights. For all tests, *p*-values ≤ 0.05 were considered to indicate statistical significance. Data analysis was performed using Stata version 14 [24].

## 3. Results

### 3.1. Sample Characteristics

The study sample comprised 101,408 men and 678,460 women aged 15–49 years. Table 1 presents the sample distribution by substance use pattern and other covariates used in the study. Nearly half (47%) of the male participants were non-users of tobacco and alcohol, and most of the female participants were not using any substances (91.6%). Among the eligible participants, 7% of men and 2% of women used to smoke only, whereas 13.5% of men and 5% of women were exclusive smokeless tobacco users. Less than one percent of men (0.6%) and women (0.1%) were daily consumers of only alcohol, while nearly 8% of men and 0.7% of women were exclusively irregular consumers of alcohol. 

### 3.2. Prevalence of Hypertension and Bivariate Association

Overall, 13.5% of men and 8.3% of women were found with hypertensive status (Table 2). The prevalence of hypertension was observed to be the highest among those men who were daily consumers of alcohol and SLT (27.1%), followed by those who were daily consumers of alcohol only (26%). Similarly, the prevalence of hypertension was estimated the highest among those women who were the daily consumers of alcohol along with using both forms of tobacco (21.6%), followed by the women having alcohol daily along with SLT (20.7%).

Prevalence of hypertension was noted to increase with advancing age. Single males (6.3%) and females (2.9%) were recorded with the least prevalence of hypertension, while the highest prevalence was noted among the widowed/separated/divorced (males: 18.9%; females: 14%). The prevalence of hypertension in women was higher among those with a lower level of formal education. On the other hand, there was a marginal difference in the prevalence of hypertensive cases among men with different levels of formal education.

The prevalence of hypertension was higher among the wealthier group of the population. The observed difference in the prevalence between the richer and the richest wealth quintile among the female population was negligible (with an overlapping 95% confidence interval). Urban males (14.2%) had a higher prevalence of hypertension compared to their rural (12.1%) counterparts. However, the urban–rural difference in the prevalence of hypertension was marginal among the female population. Northeastern states had the highest prevalence of hypertension.

### 3.3. Mixed Effect of Substance Use on Hypertension: Regression Analysis

Multivariate analysis revealed higher odds of hypertension among men (OR: 1.85; 95% CI: 1.60–2.14) who consumed all the three substances (including daily consumption of alcohol) compared to non-users (Figure 2). However, the men who consumed alcohol daily along with the SLT use had recorded the highest odds (OR: 2.32; 95% CI: 1.99–2.71) for being hypertensive compared to non-users.

Note: Adjusted odds ratio accounted for age, marital status, education, body mass index, religion, social group, occupation, wealth quintile, place of residence (urban/rural), and region of residence (Indian states).

Male and female smokers consuming alcohol daily had 73% (OR: 1.73; 95% CI: 1.49–2.01) and 71% (OR: 1.71; 95% CI: 1.14–2.56) more chances of being hypertensive than no-substance-users, respectively. Exclusive consumption of alcohol had a higher likelihood of getting hypertension than tobacco consumption. However, women smokers (OR: 1.16; 95% CI: 1.09–1.23) and men SLT users (OR: 1.10; 95% CI: 1.03–1.18) were also found relatively more likely to be hypertensive than no-substance-users.

Daily consumption of alcohol was noted to increase the chance of hypertension significantly among both men (OR: 1.55; 95% CI: 1.28–1.86) and women (OR: 1.36; 95% CI: 1.15–1.60). Even the irregular (not daily) consumption of alcohol along with SLT use, had 54% and 37% higher likelihood of getting hypertension among men (OR: 1.54; 95% CI: 1.43–1.66) and women (OR: 1.37; 95% CI: 1.27–1.48) compared to no-substance-users, respectively. Male and female smokers with irregular (not daily) consumption of alcohol were 12% (OR: 1.12; 95% CI: 1.04–1.21) and 48% (OR: 1.48; 95% CI: 1.15–1.9) more likely to having hypertension compared to no-substance-users, respectively.

Moreover, the increasing age, lower education and obesity were the significantly contributing factors in the likelihood of getting hypertensive (Table A1). Interestingly, women from the lowest economic stratum (OR: 1.14; 95% CI: 1.09–1.19) were more likely to be hypertensive compared to the highest economic stratum, while men from the poorest wealth quintile had 17% less chance of getting hypertension than those from the richest wealth quintile.

## 4. Discussion

This study helps understand the complex association between patterns of multiple substance use and hypertension using the nationally representative sample from a socio-culturally diverse country, such as India. Findings substantiated that tobacco and alcohol consumption is detrimental to health by increasing the chances of hypertension manifolds for both men and women. Daily consumption of alcohol and SLT among men and women were found with the highest likelihood of getting hypertension compared to the no-substance-use. Daily consumption of alcohol substantially aggravates the vulnerability among tobacco users for being hypertensive. Alcohol and tobacco intake had been established as strong risk factors for hypertension [25]; however, providing the separate and combined effects of different substance use is new to the contemporary pool of evidence. Men consuming alcohol daily with SLT were found over two-fold more likely of being hypertensive compared to no-substance-users. The finding seconds the evidence recorded in the previous study [26], where the higher prevalence of hypertension was noted among males and females consuming locally produced alcohol and chewing tobacco. The exclusive association of smoking among women with hypertension was more elevated and statistically significant than among men compared to no-substance-users. This is also noted in previous studies in different regions of the world [27,28]. A recent population-based cohort study found female smokers at higher risk than their male counterparts for elevated systolic blood pressure when increasing their daily cigarette consumption [4]. Current findings, along with evidence in the literature, highlights the need for customizing gender-sensitive nicotine cessation programs as epidemiological studies have noted differentiating sensory impact of cigarettes among women than men, which should further be explored for the Indian population [29]. Tobacco control is to be made more effective by improving awareness about its associated health risks reaching the grassroots level and promoting capacity building and outreach under the mandate of the WHO Framework Convention on Tobacco Control and National Tobacco Control Programme [30].

Individuals consuming alcohol daily even without consuming any form of tobacco were at a higher risk of being hypertensive than the no-substance-users. Alcohol control policies could consider WHO’s “five best buys” interventions for preventing and controlling non-communicable diseases; for instance, setting the number of standard drinks or stringent rules for on- and off-premise sales should be established [31].

Males and females from higher age group cohorts have been recorded with higher systolic and diastolic blood pressure [32]. Sudharsanan and Geldsetzer [33] highlight the rapidly ageing population of a developing economy, such as India, to account for about 93% of the disease momentum of hypertension. This is predicted to overwhelm the current non-communicable diseases control program, and thus there is an urgent need for expanding healthcare services promoting primordial prevention. This study has also found females belonging to the poorest household with the highest likelihood of getting hypertension, contrary to the existing evidence [34]. However, it should be noted while drawing implications that the current study excluded people being treated or currently on medication for hypertension.

The scope of evaluation by the current study was limited in establishing the association between variables due to the cross-sectional nature of the data collected by the survey. The prevalence of chronic morbidities, such as hypertension, is more pronounced in advanced ages; however, the study’s findings are restricted to 49 years for both sexes. The survey considers self-reported tobacco consumed by the participant in the last 24 h, and thus the likelihood of getting hypertension among long-term users could not be extrapolated from the dataset made available by the survey. The information on the frequency of tobacco use among the participants was also not available in the survey.

## 5. Conclusions

In order to develop an evidence-based and more holistic tobacco and alcohol control program in countries, such as India, there is a need to explore all possible nuances in the pattern of substance use among different population groups. Tobacco and alcohol consumption are primary risk factors for hypertension and enhance the vulnerability for other non-communicable diseases. This study attempted to bridge the gap in the literature by examining the effect of mixed/combined use of alcohol and smoking and smokeless tobacco on hypertension among the adult population. The findings highlighted the apparent contrasts in the prevalence of hypertension among both sexes having a similar pattern of substance use. This study recommends that policymakers at the central and state level use the current study’s findings to guide the scaling up of primordial and primary prevention along with existing cardiac rehabilitation programs that are more sensitive to specific combination groups and socioeconomic strata, especially the sex of the population. The focused interventions would bring affordable healthcare management within reach of communities lacking the resources to manage the compounded effects of such chronic morbidities.

## Figures and Tables

**Figure 1 ijerph-19-03239-f001:**
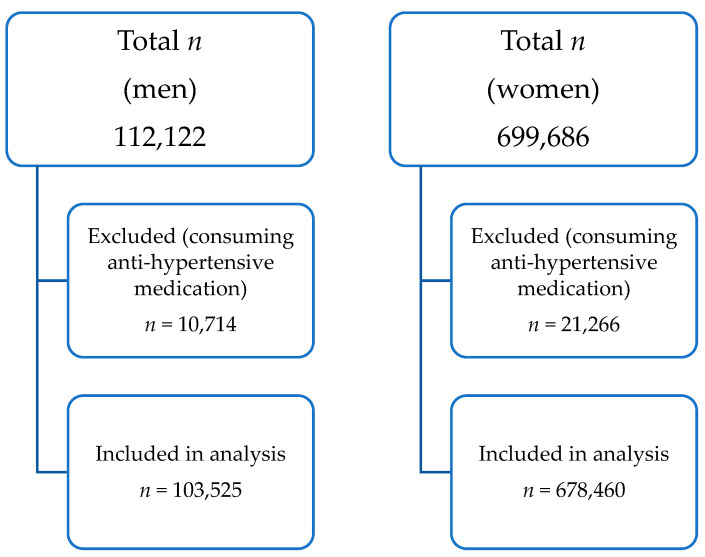
Sample flow of the study population.

**Figure 2 ijerph-19-03239-f002:**
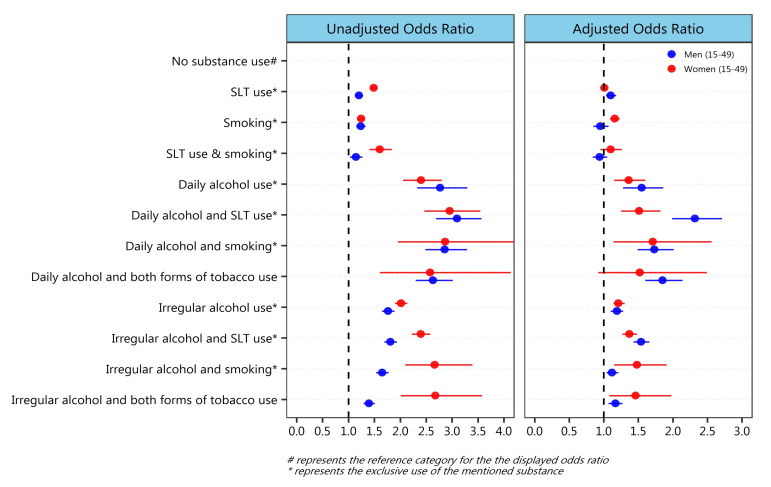
Unadjusted and adjusted effect (odds ratio) of mutually exclusive and mixed consumption patterns of alcohol and tobacco among men and women aged 15–49 on their hypertensive status, India, 2015–2016.

**Table 1 ijerph-19-03239-t001:** Sample distribution of men and women aged 15–49 years by key predictor and select correlates, India, 2015–2016.

	Men (15–49 Years)		Women (15–49 Years)	
	% (95% CI)	*n*	% (95% CI)	*n*
**Substance use**				
No substance use	47.00 (46.53, 47.47)	44,546	91.57 (91.47, 91.66)	594,297
SLT use *	13.54 (13.23, 13.85)	13,593	5.04 (4.97, 5.11)	51,814
Smoking *	7.41 (7.16, 7.67)	8187	2.03 (1.97, 2.09)	13,603
SLT use and smoking *	2.97 (2.83, 3.12)	3572	0.13 (0.12, 0.15)	1972
Daily alcohol use *	0.64 (0.58, 0.72)	716	0.14 (0.13, 0.15)	1139
Daily alcohol and SLT use *	0.80 (0.73, 0.87)	1044	0.06 (0.05, 0.07)	722
Daily alcohol and smoking *	1.09 (1.00, 1.18)	1115	0.01 (0.01, 0.02)	166
Daily alcohol and both forms of tobacco use *	0.82 (0.75, 0.89)	1240	0.01 (0.01, 0.01)	116
Irregular alcohol use *	7.87 (7.59, 8.15)	7110	0.65 (0.63, 0.68)	8917
Irregular alcohol and SLT use *	6.38 (6.18, 6.58)	7430	0.30 (0.29, 0.32)	4974
Irregular alcohol use and smoking *	6.95 (6.70, 7.22)	6738	0.04 (0.03, 0.04)	440
Irregular alcohol and both forms of tobacco	4.53 (4.36, 4.70)	6117	0.02 (0.01, 0.02)	300
**Age group**				
15–19	18.36 (18.01, 18.72)	18,939	17.75 (17.62, 17.88)	123,627
20–24	16.27 (15.92, 16.62)	16,475	17.87 (17.73, 18.01)	121,159
25–29	15.76 (15.42, 16.12)	15,917	16.64 (16.51, 16.78)	112,805
30–34	14.17 (13.85, 14.51)	14,374	13.89 (13.76, 14.01)	94,391
35–39	13.31 (12.98, 13.64)	13,573	12.92 (12.79, 13.04)	87,003
40–44	11.41 (11.12, 11.71)	11,506	10.84 (10.73, 10.95)	72,449
45–49	10.72 (10.43, 11.01)	10,624	10.10 (9.99, 10.20)	67,026
**Marital status**				
Never in union	38.76 (38.30, 39.22)	39,729	23.20 (23.05, 23.35)	169,845
Currently married	60.01 (59.54, 60.47)	60,410	72.69 (72.53, 72.85)	481,722
Widowed/divorced/separated	1.23 (1.12, 1.35)	1269	4.10 (4.03, 4.18)	26,893
**Education**				
Higher	17.66 (17.27, 18.05)	16,055	12.82 (12.69, 12.95)	77,840
Secondary	58.47 (58.00, 58.93)	60,548	47.42 (47.24, 47.60)	325,515
Primary	12.01 (11.72, 12.31)	12,438	12.40 (12.28, 12.52)	85,115
No education	11.86 (11.58, 12.15)	12,367	27.36 (27.21, 27.51)	189,990
**Body Mass Index**				
Normal	45.62 (45.15, 46.09)	47,477	44.82 (44.64, 44.99)	315,665
Underweight	20.43 (20.07, 20.80)	19,450	22.80 (22.66, 22.95)	148,146
Overweight	26.71 (26.28, 27.15)	25,178	22.75 (22.59, 22.91)	147,066
Obese	7.24 (6.96, 7.53)	6140	9.63 (9.51, 9.76)	55,117
**Religion**				
Hindu	81.41 (81.04, 81.77)	75,553	80.68 (80.53, 80.82)	504,708
Muslim	13.35 (13.03, 13.68)	14,135	13.72 (13.59, 13.85)	90,857
Others	5.23 (5.04, 5.44)	11,720	5.6 (5.52, 5.69)	82,895
**Social group**				
Others	24.58 (24.13, 25.04)	20,427	23.98 (23.81, 24.15)	139,358
OBC	45.44 (44.97, 45.91)	39,376	45.15 (44.97, 45.33)	265,838
SCs	20.71 (20.33, 21.10)	18,139	21.24 (21.09, 21.40)	121,350
STs	9.26 (9.04, 9.49)	18,080	9.62 (9.53, 9.72)	124,040
**Occupation**				
Professional/technical/managerial	6.32 (6.06, 6.59)	5567	0.50 (0.48, 0.53)	3352
Clerical	1.97 (1.83, 2.12)	1772	0.07 (0.06, 0.09)	456
Sales	9.33 (9.05, 9.63)	8831	0.25 (0.23, 0.27)	1806
Agricultural	25.92 (25.55, 26.30)	28,678	2.55 (2.50, 2.60)	18,028
Services/household and domestic	6.95 (6.69, 7.21)	6908	0.58 (0.56, 0.61)	3854
Manual—skilled and unskilled	26.10 (25.68, 26.53)	25,612	1.09 (1.05, 1.13)	7338
Not in work force/no occupation	23.41 (23.02, 23.81)	24,040	94.95 (94.87, 95.02)	643,626
**Wealth quintile**				
Richest	22.81 (22.37, 23.26)	20,161	20.77 (20.61, 20.93)	125,710
Richer	22.07 (21.66, 22.48)	20,955	21.04 (20.88, 21.19)	133,512
Middle	21.39 (21.02, 21.76)	22,170	20.60 (20.46, 20.74)	142,783
Poorer	18.90 (18.56, 19.24)	21,288	19.67 (19.54, 19.80)	145,735
Poorest	14.84 (14.55, 15.12)	16,834	17.92 (17.80, 18.04)	130,720
**Place of residence**				
Urban	38.08 (37.58, 38.58)	31,949	34.41 (34.22, 34.6)	197,288
Rural	61.92 (61.42, 62.42)	69,459	65.59 (65.4, 65.78)	481,172
**India**	100	101,408	100	678,460

CI = confidence interval; SLT = smokeless tobacco; OBC = other backward class; SC = scheduled caste; ST = scheduled tribe. * Represents the exclusive use of the mentioned substance. *n* represents unweighted sample size.

**Table 2 ijerph-19-03239-t002:** Prevalence of hypertension among Indian men and women aged 15–49 years by key predictor and select correlates, India, 2015–2016.

	Men (15–49 Years)		Women (15–49 Years)	
	% (95% CI)	*p*	% (95% CI)	*p*
**Substance use**		<0.001		<0.001
No substance use	10.50 (10.06, 10.95)		7.95 (7.85, 8.05)	
SLT use *	12.78 (11.97, 13.62)		11.85 (11.40, 12.31)	
Smoking *	12.45 (11.34, 13.65)		9.19 (8.41, 10.03)	
SLT use and smoking *	11.53 (10.07, 13.17)		13.75 (11.03, 17.02)	
Daily alcohol use *	26.02 (21.10, 31.63)		15.03 (12.04, 18.60)	
Daily alcohol and SLT use *	27.09 (23.30, 31.25)		20.72 (15.30, 27.43)	
Daily alcohol and smoking *	24.15 (20.69, 27.99)		19.06 (10.92, 31.15)	
Daily alcohol and both forms of tobacco use *	21.98 (18.43, 26.00)		21.60 (13.13, 33.43)	
Irregular alcohol use *	16.56 (15.11, 18.12)		13.60 (12.23, 15.09)	
Irregular alcohol and SLT use *	17.60 (16.33, 18.95)		18.09 (16.38, 19.93)	
Irregular alcohol use and smoking *	15.68 (14.34, 17.13)		14.04 (9.82, 19.67)	
Irregular alcohol and both forms of tobacco	13.61 (12.25, 15.10)		18.35 (12.42, 26.25)	
**Age group**		<0.001		<0.001
15–19	2.96 (2.63, 3.33)		2.09 (1.98, 2.20)	
20–24	6.63 (6.12, 7.18)		3.25 (3.11, 3.40)	
25–29	10.48 (9.73, 11.28)		5.19 (5.00, 5.39)	
30–34	14.31 (13.46, 15.21)		8.36 (8.10, 8.63)	
35–39	18.66 (17.57, 19.80)		12.11 (11.78, 12.45)	
40–44	22.17 (20.99, 23.39)		16.05 (15.65, 16.46)	
45–49	24.01 (22.82, 25.24)		19.66 (19.21, 20.11)	
**Marital status**		<0.001		<0.001
Never in union	6.32 (5.95, 6.72)		2.91 (2.79, 3.03)	
Currently married	16.98 (16.52, 17.45)		9.64 (9.52, 9.77)	
Widowed/divorced/separated	18.94 (15.48, 22.97)		13.99 (13.37, 14.64)	
**Education**		<0.001		<0.001
Higher	13.78 (12.95, 14.67)		5.47 (5.23, 5.73)	
Secondary	12.09 (11.69, 12.50)		6.83 (6.70, 6.97)	
Primary	14.85 (13.84, 15.92)		10.06 (9.76, 10.37)	
No education	13.65 (12.80, 14.54)		11.22 (11.03, 11.42)	
**Body Mass Index**		<0.001		<0.001
Normal	9.85 (9.46, 10.26)		6.07 (5.95, 6.19)	
Underweight	5.17 (4.76, 5.61)		4.21 (4.08, 4.35)	
Overweight	19.4 (18.64, 20.18)		12.13 (11.88, 12.38)	
Obese	29.69 (27.86, 31.58)		19.05 (18.55, 19.56)	
**Religion**		<0.001		<0.001
Hindu	12.91 (12.55, 13.27)		8.01 (7.90, 8.12)	
Muslim	11.44 (10.64, 12.29)		8.98 (8.69, 9.26)	
Others	16.53 (15.19, 17.96)		10.38 (9.93, 10.84)	
**Social group**		<0.001		<0.001
Others	14.09 (13.32, 14.90)		9.04 (8.81, 9.28)	
OBC	12.37 (11.92, 12.84)		7.81 (7.67, 7.95)	
SCs	12.66 (11.95, 13.40)		7.80 (7.59, 8.02)	
STs	13.46 (12.61, 14.35)		9.00 (8.73, 9.28)	
**Occupation**		<0.001		<0.001
Professional/technical/managerial	18.67 (17.02, 20.44)		6.57 (5.45, 7.91)	
Clerical	17.20 (14.49, 20.28)		8.15 (5.29, 12.35)	
Sales	16.81 (15.57, 18.12)		9.45 (7.39, 12.02)	
Agricultural	13.37 (12.83, 13.92)		9.13 (8.53, 9.76)	
Services/household and domestic	15.99 (14.61, 17.47)		9.31 (7.96, 10.87)	
Manual—skilled and unskilled	14.11 (13.43, 14.83)		8.79 (7.88, 9.78)	
Not in work force/no occupation	6.69 (6.24, 7.18)		8.24 (8.14, 8.34)	
**Wealth quintile**		<0.001		<0.001
Richest	15.23 (14.44, 16.05)		8.91 (8.66, 9.16)	
Richer	15.07 (14.26, 15.91)		9.11 (8.87, 9.36)	
Middle	12.90 (12.25, 13.57)		7.98 (7.77, 8.19)	
Poorer	10.49 (9.93, 11.08)		7.66 (7.47, 7.85)	
Poorest	9.41 (8.84, 10.01)		7.60 (7.42, 7.79)	
**Place of residence**		<0.001		<0.001
Urban	14.23 (13.58, 14.91)		8.75 (8.55, 8.97)	
Rural	12.12 (11.79, 12.46)		8.03 (7.92, 8.13)	
**Region**		<0.001		<0.001
North	16.82 (16.05, 17.62)		9.70 (9.44, 9.97)	
Central	10.70 (10.26, 11.15)		7.47 (7.33, 7.61)	
East	11.55 (10.85, 12.30)		7.63 (7.42, 7.85)	
West	13.93 (13.10, 14.79)		8.29 (8.02, 8.57)	
Northeast	18.71 (17.71, 19.76)		13.72 (13.38, 14.07)	
South	15.09 (14.32, 15.89)		8.40 (8.16, 8.64)	
**India**	**13.46 (13.14, 13.79)**		**8.27 (8.17, 8.37)**	

CI = confidence interval; SLT = smokeless tobacco; OBC = other backward class; SC = scheduled caste; ST = scheduled tribe. * Represents the exclusive use of the mentioned substance. *p* < 0.05 identifies statistical significance, determined with the Pearson Chi-Squared test for % differences.

## Data Availability

Data available through the DHS Online Portal https://dhsprogram.com/data/ (accessed on 22 January 2021).

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
