# Peer review of "Mixed Effect of Alcohol, Smoking, and Smokeless Tobacco Use on Hypertension among Adult Population in India: A Nationally Representative Cross-Sectional Study"

_ijerph, 2022, doi:10.3390/ijerph19063239_

Round 1
Reviewer 1 Report
Introduction:
- Provide prevalence over time of alcohol and tobacco consumption in India.
Methods:
- Briefly explain why there is such large difference in the number of women and men interviewed. Regardless of whether each sub-sample is representative.
- Please provide reference of other studies when using the threshold.
- The conceptual framework should be explicitly introduced in the introduction to understand why this frame and not other was chosen.
Results
- Figure 2 is not necessary since these results are not used in the discussion. It would be informative if a semiadjusted odds ratio results would be introduced when controlling for regional differences only.
Discussion
- Please change novel to importante in the first sentence of the discussion.
- The discussion of regional difference could be introduced if adjusted variable were introduced in Figure 2, therefore I suggest to avoid any discussion of regional differences.
- Authors should explicitly use the theoretical frame they claim they used (WHO frame) to explain null differences between men and women. This is the key of manuscript
Author Response
Manuscript ID: ijerph-1563249 entitled "Mixed effect of alcohol, smoking and smokeless tobacco use on hypertension among adult population in India: a nationally representative cross-sectional study”.
Dear Editorial Team,
I am grateful to the reviewers for their comments and thank you for giving us a chance to revise our manuscript titled "Mixed effect of alcohol, smoking and smokeless tobacco use on hypertension among adult population in India: a nationally representative cross-sectional study” (: ijerph-1563249). The changes have been made in the revised manuscript. Please see our responses to the reviewers' comments as follows:
Reviewer 1
Introduction:
- Provide prevalence over time of alcohol and tobacco consumption in India.
Response: Thank you for this comment. The introduction has been revised accordingly.
Methods:
- Briefly explain why there is such large difference in the number of women and men interviewed. Regardless of whether each sub-sample is representative.
Response: The structure of Indian DHS aims to collect data about women and child health from reproductive aged women (15-49 years) and in order to ensure that the sample is representative at the national as well as the sub-regional levels, more women than men were considered in the survey.
- Please provide reference of other studies when using the threshold.
Response: The data collected by the survey does not measure the intake of tobacco and alcohol with regards to a mandated threshold and focuses on assessing the magnitude and frequency of different forms instead hence the study objective did not include measures disaggregated in accordance with specific threshold.
- The conceptual framework should be explicitly introduced in the introduction to understand why this frame and not other was chosen.
Response: Thank you for highlighting this concern, the same has been updated accordingly.
Results
- Figure 2 is not necessary since these results are not used in the discussion. It would be informative if a semi-adjusted odds ratio results would be introduced when controlling for regional differences only.
Response: Thank you for the suggestion. We have removed the figure and edited the manuscript accordingly.
Discussion
- Please change novel to important in the first sentence of the discussion.
Response: Thank you for the highlighting the oversight, it has been updated.
- The discussion of regional difference could be introduced if adjusted variable were introduced in Figure 2, therefore I suggest to avoid any discussion of regional differences.
Response: Thank you for the highlighting the oversight, it has been updated.
- Authors should explicitly use the theoretical frame they claim they used (WHO frame) to explain null differences between men and women. This is the key of manuscript
Response: Thank you for the highlighting the oversight, it has been updated.
Reviewer 2 Report
This study assesses the mutually exclusive and mixed consumption patterns of alcohol; tobacco smoking and smokeless tobacco use and their association with hypertension among adult population in India.
The introduction does not contain any information related to the health profile of the Indian population.
Appropriately present the limitations of the study as well as the practical utility of the reported results.
I suggest that the authors could benefit from the results and conclusion of the following recent studies to be included in the discussion: https://doi.org/10.3390/ijerph17228565 , https://doi.org/10.3390/jcm9082612
I encourage the authors to expand the discussion section.
The conclusion needs improvement.
I strongly recommend the authors seek English language revision for this manuscript. I believe this would help clarify some of the expressions and sentences that are currently not appropriate or incomprehensible.
Author Response
Manuscript ID: ijerph-1563249 entitled "Mixed effect of alcohol, smoking and smokeless tobacco use on hypertension among adult population in India: a nationally representative cross-sectional study”.
Dear Editorial Team,
I am grateful to the reviewers for their comments and thank you for giving us a chance to revise our manuscript titled "Mixed effect of alcohol, smoking and smokeless tobacco use on hypertension among adult population in India: a nationally representative cross-sectional study” (: ijerph-1563249). The changes have been made in the revised manuscript. Please see our responses to the reviewers' comments as follows:
Reviewer 2
This study assesses the mutually exclusive and mixed consumption patterns of alcohol; tobacco smoking and smokeless tobacco use and their association with hypertension among adult population in India.
- The introduction does not contain any information related to the health profile of the Indian population.
Response: Thank you for the highlighting the oversight, it has been updated.
- Appropriately present the limitations of the study as well as the practical utility of the reported results.
Response: Thank you for the highlighting the oversight, it has been updated.
- I suggest that the authors could benefit from the results and conclusion of the following recent studies to be included in the discussion: https://doi.org/10.3390/ijerph17228565 ,https://doi.org/10.3390/jcm9082612
Response: Thank you for providing a valuable input about the research article, the inputs has been updated.
- I encourage the authors to expand the discussion section.
Response: Thank you for the comment. The discussion has been updated with edits related to regional variation observed and
The conclusion needs improvement.
Response: Thank you for your comment. Conclusion has been edited accordingly..
- I strongly recommend the authors seek English language revision for this manuscript. I believe this would help clarify some of the expressions and sentences that are currently not appropriate or incomprehensible.
Response: Thank you for the comment. The manuscript has been revised accordingly.
Round 2
Reviewer 1 Report
Thank you for answering my suggestions and concerns
Author Response
Thank you for providing your constructive comments and suggestions which improved the manuscript significantly.
Reviewer 2 Report
No comments and suggestions for the revised manuscript.
Author Response

(The authors gave the same response as above.)
